# RETHINKING CLIP FOR LONG-TAILED CLASS-INCREMENTAL LEARNING

## ABSTRACT

Pre-trained vision–language models such as CLIP provide strong priors for class-incremental learning (CIL), yet existing methods degrade sharply in long-tailed scenarios. We demonstrate that CLIP, with only a single lightweight adapter, is sufficient to handle this setting when the CIL process is structured into Intra-Task Stabilization, Inter-Task Preservation, and Knowledge Consolidation. In Intra-Task Stabilization, we ground the training of current tail classes through Two-Stage Hybrid Augmentation, which anchors learning on CLIP's text knowledge and refines it with distribution-aware signals. In Inter-Task Preservation, we protect past knowledge with Tail-Aware Semantic Shrinkage, which corrects biased statistics using semantically related head classes, and Adaptive Margin Hinge Loss, which maintains robust boundaries between old and new classes. Finally, in Knowledge Consolidation, Mode-Connectivity Spherical Merge integrates task-specific adapters along a low-loss path, ensuring stable unification into a single model. By explicitly linking Intra-Task Stabilization, Inter-Task Preservation, and Knowledge Consolidation, our framework delivers a coherent solution for long-tailed CIL. Experiments on ImageNetSubset with increasing class numbers show consistent improvements over prior CLIP-based methods, with margins that grow under more severe long-tail conditions.

## 1 INTRODUCTION

Class incremental learning (CIL) has been a long-standing problem in AI toward lifelong learning of streaming information. However, CIL is challenged by catastrophic forgetting (McCloskey & Cohen, 1989), where knowledge of past tasks is erased. While methods like regularization (Kirkpatrick et al., 2017) and replay (Rebuffi et al., 2017) have been proposed, most research operates under the idealized assumption of balanced data. In reality, data often follows a long-tailed distribution (Liu et al., 2019; Kang et al., 2020), a condition that worsen forgetting for rare, tail classes.

The advent of models like CLIP (Radford et al., 2021) has revolutionized CIL, offering strong baseline resistance to forgetting (Thengane et al., 2022). Subsequent works have applied parameter-efficient tuning techniques like adapters (Luo et al., 2025) or prompts (Wang et al., 2022b). However, a critical blind spot remains: their effectiveness is shown almost exclusively on balanced benchmarks. When applied to long-tailed scenarios, their performance degrades, suggesting that existing methods fail to harness CLIP's rich prior knowledge for data-scarce classes properly. This motivates our conjecture that a more effective strategy is needed to harness CLIP's pre-trained knowledge.

In this paper, we present a novel framework designed for the challenging long-tailed, exemplar-free CIL setting, aiming to fully leverage the rich information embedded in pre-trained CLIP. Our approach employs a single, task-shared adapter that is significantly lighter than the add-ons used in other CLIP-based CIL methods, enabling efficient knowledge accumulation over time. We show that this minimal add-on, combined with proper extraction of CLIP representations, is sufficient to tackle the problem. To effectively leverage CLIP representations, we propose a framework that systematically tackles the CIL lifecycle in three logical stages: Intra-Task Stabilization using Two-Stage Hybrid Augmentation for robust tail-class feature augmentation; Inter-Task Preservation using Tail-aware Semantic Shrinkage and Adaptive Margin Hinge Loss to correct tail-class statistics and refine decision boundaries; and Knowledge Consolidation using Mode-Connectivity Spherical Merge for stable integration of knowledge into the single adapter.

We conduct extensive experiments on the challenging long-tailed ImageNetSubset and CIFAR-100 benchmarks (Liu et al., 2022), demonstrating that our method significantly outperforms existing state-of-the-art approaches. Notably, our proposed method, which employs a minimal adapter module maximizing the use of CLIP's pre-trained knowledge, achieves greater performance gains over other CLIP-based CIL approaches as the number of classes increases, resulting in more severe long-tailed conditions. Our contributions can be summarized as follows.

- To the best of our knowledge, we first conducted an in-depth study of modern CLIP-based CIL methods in realistic long-tailed settings, revealing the need to maximize knowledge utilization rather than simply integrating complex add-on modules.

- We propose a novel, exemplar-free framework with a single lightweight adapter that outperforms state-of-the-art methods. It features four synergistic components: Two-Stage Hybrid Augmentation, Tail-aware Semantic Shrinkage, Adaptive Margin Hinge Loss, and Mode-Connectivity Spherical Merge.

- Our method shows improved robustness in scaled-up long-tailed CIL scenarios, highlighting our conjecture that the key lies in maximizing the utilization of CLIP pre-trained knowledge rather than relying on complex add-ons without careful information manipulation.

## 2 RELATED WORK

### 2.1 CLASS-INCREMENTAL LEARNING

The primary objective of class-incremental learning (CIL) is to enable a model to learn new classes sequentially without compromising its performance on previously learned ones. The fundamental obstacle in this task is "catastrophic forgetting," a phenomenon where a neural network's knowledge of past tasks is abruptly and almost completely lost upon learning a new one (McCloskey & Cohen, 1989). Early studies aimed at mitigating this issue primarily focused on two main strategies: regularization-based and replay-based methods. Regularization-based approaches, which are typically exemplar-free, impose constraints on the learning process to protect consolidated knowledge. These methods include penalizing changes to important weights in the parameter space (*e.g.*, EWC (Kirkpatrick et al., 2017)) and using knowledge distillation to preserve the model's output on past classes (*e.g.*, LwF (Li & Hoiem, 2016)). In contrast, replay-based methods directly approach forgetting by storing a small subset of past data, or "exemplars," and replaying them during training (Rebuffi et al., 2017; Castro et al., 2018; Wu et al., 2019; Shin et al., 2017). However, this approach introduces challenges related to memory, computational cost, and data privacy.

### 2.2 CIL WITH CLIP

The advent of large-scale pre-trained vision-language models, particularly CLIP (Radford et al., 2021), has marked a paradigm shift in CIL. The rich representations learned by CLIP provide a strong inherent resistance to catastrophic forgetting, with a frozen CLIP model acting as a powerful zero-shot baseline (Thengane et al., 2022). Recent subsequent research has focused on parameter-efficient finetuning techniques. These include classifier-centric approaches that adjust the classification head using text-image semantics (Huang et al., 2024), and representation-learning approaches that modify visual features by introducing lightweight modules, such as adapters or projections (Zhou et al., 2025; Yu et al., 2024; Jha et al., 2024). However, these prior approaches share a critical limitation: they are predominantly evaluated on balanced datasets, such as splits of CIFAR-100 (Krizhevsky et al., 2009) or ImageNet (Russakovsky et al., 2015). They implicitly assume a uniform class distribution, a scenario that rarely reflects real-world data. This raises a crucial question: *can these methods effectively leverage CLIP's potential when faced with the severe data imbalance of a long-tailed distribution?* Our work addresses this research gap, arguing that a new set of techniques focused on feature enhancement, not just adaptation, is necessary.

Our research tackles the challenging intersection of long-tailed distributions, class-incremental learning, and an exemplar-free setting. Inspired by the decoupling principle in long-tail recognition (Kang et al., 2020), our architecture pairs a frozen CLIP backbone with a lightweight trainable adapter. While existing solutions for this setting rely on complex dynamic architectures (Wang et al., 2022a) or costly generative replay (Shin et al., 2017), we build upon the more efficient statistical re-

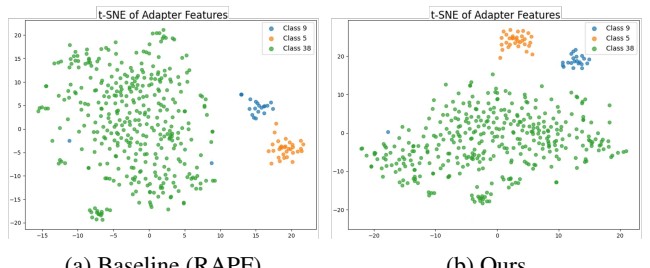

| Class | RAPF | Ours |
|---|---|---|
| 5 | 0.0010 | **0.0007** |
| 9 | 0.0011 | **0.0009** |
| 38 | 0.0010 | **0.0007** |

Intra-class variance ($\downarrow$ lower is better).

(a) Baseline (RAPF)  (b) Ours

Figure 1: t-SNE visualization of adapter features for two tail classes (5, 9) and one head class (36), along with their intra-class variances.

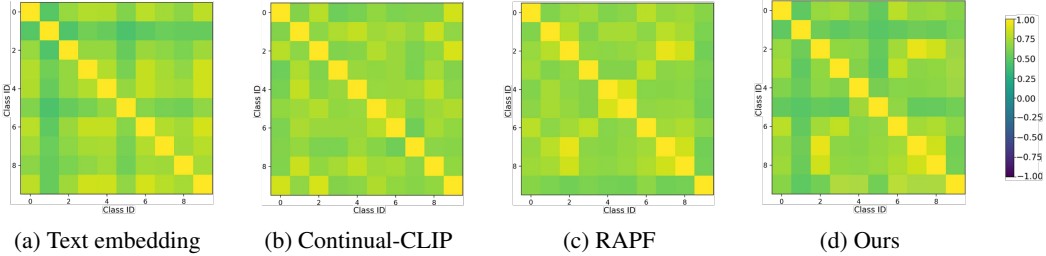

(a) Text embedding  (b) Continual-CLIP  (c) RAPF  (d) Ours

Figure 2: Visualization of inter-class semantic similarity structures. The similarity is measured between class prototypes after the final task. (a) The ground truth structure from CLIP's text embeddings. (b, c) Baselines. (d) Our method. Our method's feature space (d) most closely resembles the ground truth structure (a).

play approach (Huang et al., 2024; Hayes & Kanan, 2020; Tang et al., 2023; Zhang et al., 2023), proposing a more advanced form.

Our research tackles the challenging intersection of long-tailed distributions, class-incremental learning, and an exemplar-free setting.Inspired by the decoupling principle in long-tail recognition (Kang et al., 2020), our architecture pairs a frozen CLIP backbone with a lightweight trainable adapter. While existing solutions for this setting rely on complex dynamic architectures (Wang et al., 2022a) or costly generative replay (Shin et al., 2017), we build upon the more efficient statistical replay approach (Huang et al., 2024; Hayes & Kanan, 2020; Tang et al., 2023; Zhang et al., 2023), proposing a more advanced form.

## 3 PROPOSED METHOD

### 3.1 MOTIVATION: TWO CRITICAL FAILURES OF BASELINES IN LONG-TAILED CIL

**Failure in Intra-Task Stabilization.** The most fundamental challenge is achieving stable representations for the current task's imbalanced classes. We observe that tail classes, due to their scarcity, often yield poorly defined decision boundaries in the feature space. As visualized in the t-SNE plot in Figure 1, a representative baseline (*i.e.*, RAPF) fails at this from the outset. The features of tail classes (*e.g.*, Class 5 and 9) appear diffuse and weakly separated, revealing the model's inability to form robust decision boundaries even within a single task, a limitation we denote as intra-task stabilization failure.

**Failure in Inter-Task Preservation.** Beyond learning the initial task, the model must retain past knowledge when exposed to future tasks. However, existing methods often fail in this regard, especially when newly introduced classes are semantically similar to those previously learned. As shown in the semantic similarity matrix in Figure 2c, the feature space of a baseline method becomes visibly distorted after incremental training, losing the original semantic structure provided by CLIP's text embeddings. This demonstrates a failure of Inter-Task Preservation, leading to both catastrophic forgetting and inter-class confusion.

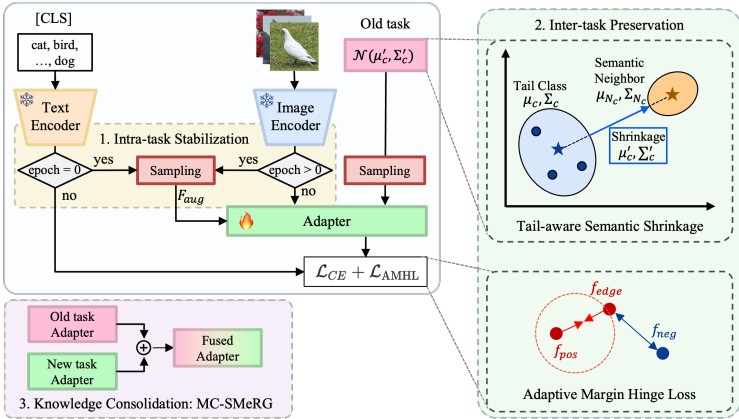

Figure 3: An overview of our proposed framework. **Top-left**: The main training pipeline for a new task. Based on the epoch, features for tail-class augmentation ($F_{aug}$) are generated either from text (epoch $= 0$) or refined image features (epoch $> 0$). **Top-right**: Tail-aware Semantic Shrinkage refines the statistics of past tail classes for replay. **Bottom-right**: Our adaptive margin loss (part of $\mathcal{L}_{AMHL}$) separates confusing class pairs. **Bottom-left**: After task training, MC-SMERG fuses the old and new adapters into a single updated adapter for future inference.

These two failures: instability within tasks and forgetting across tasks, motivate the core multi-stage design of our framework. To address these critical issues of intra-task stabilization and inter-task preservation, we introduce our framework in the following section.

## 3.2 THE PROPOSED FRAMEWORK

Figure 3 shows our proposed method, which simply utilizes a single lightweight adapter to CLIP. To systematically resolve the two issues above, we structure the CIL process into three logical stages: Intra-Task Stabilization for robust feature augmentation, Inter-Task Preservation for tail-class statistic correction and boundary refinement, and Knowledge Consolidation for stable adapter integration. These components are designed to enhance tail-class representations and ensure stable knowledge consolidation across tasks.

### 3.2.1 INTRA-TASK STABILIZATION: TWO STAGE HYBRID FEATURE SPACE AUGMENTATION

In CIL setting, when learning a new task, the model feature extractor, especially the newly added adapter, is initially in an unstable state. At this point, performing an augmentation based on the scarce real image features of tail classes carries a high risk of amplifying noise or learning a biased distribution. Conversely, after several epochs of training, once the adapter has adapted to the current task's data distribution, it becomes effective to leverage the underlying distribution of the real data. To dynamically respond to the stage-dependent changes in the model's state, we design a hybrid strategy composed of two stages: 'Zero-Shot Text-based Seeding' and 'Adaptive $k$-NN Feature Refinement'. Figure 4a and 4b provide a conceptual illustration of this two-stage process.

**Stage 1: Zero-Shot Text-based Seeding.** In the first training epoch, we bypass the yet-untrained adapter and directly leverage the vast linguistic-visual prior knowledge of the CLIP model. The goal of this stage is to generate a semantically valid initial feature space using only textual information-the class names-without any real image data. Specifically, for a given tail class $c$, we feed a text prompt (*e.g.*, *a photo of a class name*) into CLIP's text encoder to extract a text embedding $z_c$. This embedding is considered the mean $\mu_c$ of the class's feature distribution. The covariance $\Sigma_c$ is approximated by assuming an isotropic Gaussian distribution, $\Sigma_c = \sigma^2 I$. Here, the variance $\sigma^2$ is dynamically set based on the average distance between real image features and text features, ensuring the generated features have an appropriate scale. Finally, we sample virtual feature vectors from this estimated Gaussian distribution $N(\mu_c, \Sigma_c)$ to augment the initial training data. This strategy mitigates early-stage instability and steers learning toward a stable semantic center.

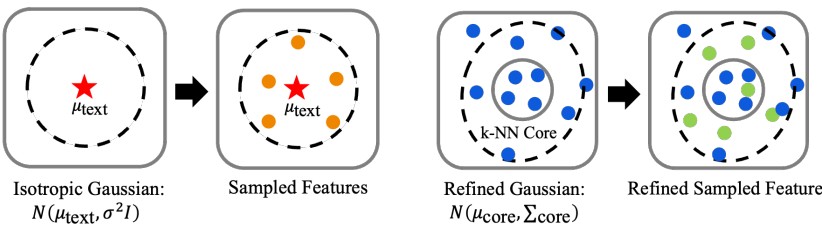

(a) Stage 1: Zero-shot text-based seeding  (b) Adaptive $k$-NN Feature refinement

Figure 4: An illustration of our Two-Stage Hybrid Feature Space Augmentation. (a) Stage 1 seeds the process by sampling features (orange) around the text embedding ($\mu_{\text{text}}$). (b) Stage 2 refines this by sampling features (green) from a robust k-NN core set (inner blue) of the real data distribution.

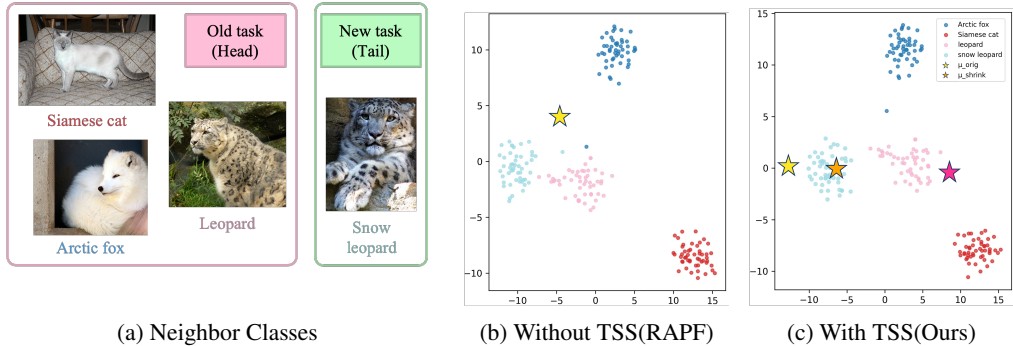

(a) Neighbor Classes  (b) Without TSS(RAPF)  (c) With TSS(Ours)

Figure 5: A t-SNE visualization of the TSS mechanism. (a) An example tail class ('Snow leopard') and its head class neighbors. (b) Without TSS, the tail class mean ($\mu_{\text{orig}}$) is biased. (c) Our TSS corrects the mean by shrinking it ($\mu_{\text{shrink}}$) towards the centroid of the neighbors (pink star), resulting in a more stable representation.

**Stage 2: Adaptive $k$-NN Feature Refinement.**   After the first epoch, the adapter is partially aligned with the current task, making real-image feature distributions more informative than abstract text priors. We therefore switch to a data-driven strategy that exploits the stabilizing adapter: rather than relying on a static, once-collected feature set, we perform augmentation using tail-class features refined through the adapter's current state. To mitigate the risk of bias from outliers, particularly problematic in classes with limited samples, we avoid using the entire feature set. Instead, we construct a robust *core* set via $k$-Nearest Neighbors ($k$-NN), ensuring that only the most representative features are retained. Concretely, within the feature set $F_c$ of each tail class $c$, the core set is formed by aggregating the $k$-NN samples for each instance. We then compute the mean $\mu_{\text{core}}$ and covariance $\Sigma_{\text{core}}$ of this core set to define a new Gaussian distribution, $N(\mu_{\text{core}}, \Sigma_{\text{core}})$, which represents the dense, region of the class. Features sampled from this refined distribution reflect a feature space that progressively approaches reality as training proceeds, thus possessing higher quality and discriminative power. Consequently, our two-stage hybrid augmentation serves as an effective strategy that secures both stability and adaptability throughout the entire learning process. The detailed algorithm of the overall process of our proposed method is summarized in supplementary material.

### 3.2.2 INTER-TASK PRESERVATION: PROTECTING AND REFINING PAST KNOWLEDGE

**Tail-aware Semantic Shrinkage.**   In exemplar-free CIL, preserving past knowledge typically relies on generating synthetic features from stored class statistics (*i.e.*, mean and covariance). While this circumvents the need to retain real images, it creates a severe vulnerability for tail classes: their statistics, derived from only a handful of samples, are noisy and unreliable. Features synthesized from such weak estimates can corrupt subsequent training, paradoxically accelerating the forgetting of the very classes intended to be preserved, as illustrated in Figure 5.

To counter this instability, we propose Tail-aware Semantic Shrinkage (TSS), a regularization strategy that refines tail-class statistics by leveraging the reliability of head classes. Specifically, TSS

*shrinks* the noisy statistics of a tail class toward the average statistics of its most semantically related head-class neighbors, guided by CLIP priors and weighted by each class's degree of *tailness*.

Specifically, the TSS process identifies $K$ semantic neighbors for a given past tail class $c$ by searching in CLIP's text embedding space. The refined statistics for class $c$, $(\mu'_c, \Sigma'_c)$, are then obtained via a convex combination of the original and the average statistics of the neighbors:

$$\mu'_c = (1 - \alpha_c)\mu_c + \alpha_c\mu_{N_c}, \quad \Sigma'_c = (1 - \alpha_c)\Sigma_c + \alpha_c\Sigma_{N_c}. \tag{1}$$

The shrinkage coefficient $\alpha_c$ is adaptively determined for each class based on both sample scarcity and statistical uncertainty. We define the uncertainty of class $c$ as $\mathcal{U}(\Sigma_c) = \mathrm{Tr}(\Sigma_c)$, the trace of its covariance matrix, which reflects the dispersion of its feature space. Using this, we compute:

$$\alpha_c = \frac{\mathcal{U}(\Sigma_c)}{\mathcal{U}(\Sigma_c) + n_c} \tag{2}$$

This formulation adaptively regularizes classes: those with high variance and limited samples are corrected more aggressively, while compact and reliable classes remain largely unaffected. The uncertainty-aware shrinkage produces more stable and semantically consistent synthetic features for replay, ultimately enhancing generalization under long-tailed distributions.

**Adaptive Margin Hinge Loss.** Another core challenge in CIL is the semantic overlap between previously learned classes and newly introduced ones. This issue is especially pronounced when a new, data-rich head class is semantically similar to a past, data-scarce tail class. With its fragile boundary, the tail class is easily overwhelmed by the strong learning signal of the head class, resulting in catastrophic forgetting and severe inter-class confusion.

To protect the feature space of vulnerable past classes, we introduce the Adaptive Margin Hinge Loss (AMHL). This loss explicitly enforces separation between confusing class pairs. For each new task, we first identify old–new class pairs with high semantic similarity by computing cosine similarity between their CLIP text embeddings. For each such pair, AMHL adopts a triplet-style objective: we synthesize an edge-like feature $f_{\text{edge}}$ by sampling from a Gaussian approximation of the old class's distribution (using its running mean and covariance). Although not exactly on the decision boundary, $f_{\text{edge}}$ serves as an ambiguous representation; AMHL then pulls $f_{\text{edge}}$ toward the positive prototype $f_{\text{pos}}$ while pushing it away from the negative prototype $f_{\text{neg}}$.

The core novelty of AMHL lies in its adaptive margin. Unlike a conventional hinge loss with a fixed margin, which ignores the severe class imbalance in CIL, AMHL dynamically sets the separation margin $\delta_c$ according to the tailness of each old class $c$ (*e.g.*, via count-based quantiles). Tail classes are assigned larger margins to enforce stronger separation, while head classes receive smaller margins to prevent over-regularization. Formally,

$$\mathcal{L}_{\text{AMHL}} = \mathbb{E}\left[\max(0, -\langle f_{\text{edge}}, f_{\text{pos}}\rangle + \langle f_{\text{edge}}, f_{\text{neg}}\rangle + \delta_c)\right],$$

where $\langle\cdot, \cdot\rangle$ denotes the dot product. By adapting the repulsive force to class vulnerability, AMHL preserves the feature space integrity of past (tail) classes without storing exemplars, which is crucial for maintaining accuracy and mitigating forgetting in long-tail CIL.

### 3.2.3 KNOWLEDGE CONSOLIDATION: MODE-CONNECTIVITY SPHERICAL MERGE

Finally, to maintain a single, efficient adapter for inference, the knowledge acquired in the new task-specific adapter $W_{\text{new}}$ should be merged into the previous adapter $W_{\text{old}}$. Various adapter fusion strategies have been proposed, ranging from simple parameter averaging (Fukuda et al., 2025), fine-grained fusion via decomposition (Huang et al., 2024), to more complex dynamic routing mechanisms (Luo et al., 2025; Qi et al., 2025). However, these approaches can be heuristic or require intricate, parameter-wise calculations.

In contrast, we propose Mode-Connectivity Spherical Merge (MC-SMERG), an efficient approach inspired by mode connectivity theory. Prior work has shown that two independently trained networks, although converging to different local minima, can often be linked by a continuous low-loss path in weight space (Draxler et al., 2018; Garipov et al., 2018; Mirzadeh et al., 2020). Rather than relying on linear interpolation—which may disrupt learned representations—we parameterize this path using Spherical Linear Interpolation (SLERP) (Shoemake, 1985). SLERP traces the shortest

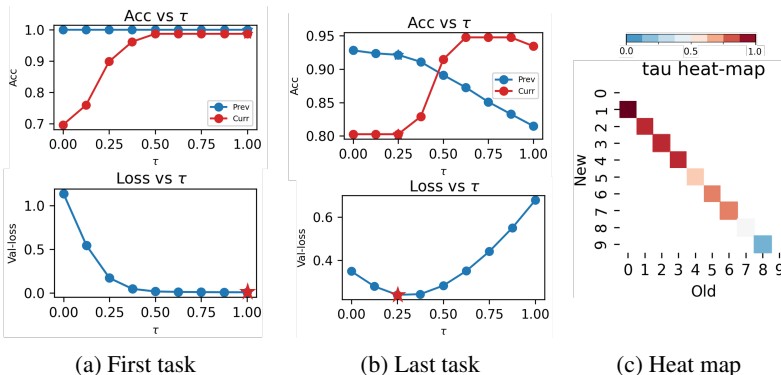

(a) First task        (b) Last task        (c) Heat map

Figure 6: Analysis of MC-SMERG dynamics in an early (Task 1) and late (Task 9) step. The accuracy plots show the stability (blue) vs. plasticity (red) trade-off. The validation loss (bottom) is minimized at an optimal $\tau^*$ (star), which shifts from a plasticity to a stability focus over time. The heatmap (right) shows this adaptive trend across all tasks.

| Methods | Exemplar | 100 Classes | | | | 200 Classes | | | | 300 Classes | | | |
|---|---|---|---|---|---|---|---|---|---|---|---|---|---|
| | | 5 | | 10 | | 5 | | 10 | | 5 | | 10 | |
| | | Last(%) | Avg(%) | Last(%) | Avg(%) | Last(%) | Avg(%) | Last(%) | Avg(%) | Last(%) | Avg(%) | Last(%) | Avg(%) |
| Continual-CLIP | ✗ | 58.60 | 64.61 | 58.60 | 67.69 | 68.34 | 75.01 | 68.34 | 76.72 | 72.31 | 78.41 | 72.31 | 67.69 |
| RAPF | ✗ | 86.12 | 88.00 | 84.02 | 89.41 | 74.62 | 79.16 | 74.10 | 81.15 | 77.10 | 80.52 | 75.84 | 82.89 |
| MoE-Adapters4CL | ✓ | 71.82 | 85.10 | 69.34 | 81.42 | 74.66 | 80.29 | 73.46 | 81.51 | 76.50 | 82.55 | 74.62 | 82.91 |
| PROOF | ✓ | 86.56 | 90.20 | 84.86 | 89.99 | 71.78 | 80.41 | 72.53 | 81.61 | 75.79 | 81.41 | 75.35 | 82.20 |
| CLAP4CLIP | ✓ | 87.25 | 90.62 | 87.19 | **91.43** | 73.36 | 79.80 | 73.92 | 80.77 | 76.30 | 82.52 | 75.08 | 83.22 |
| Ours | ✗ | **88.62** | **91.38** | **87.54** | 90.60 | **76.64** | **81.62** | **76.20** | **82.92** | **77.68** | **83.71** | **77.01** | **84.57** |

Table 1: Comparison on ImageNetSubset-LT benchmarks. '5' and '10' represent the number of tasks, and 'Last' and 'Avg' denote the last and average accuracy across tasks.

constant-speed arc on the weight hypersphere, ensuring a smooth transition between adapter states $(W_{old}, W_{new})$ while preserving parameter norms and preventing abrupt loss spikes. The resulting path $W(\tau)$ is defined as:

$$W(\tau) = \frac{\sin((1-\tau)\theta)}{\sin\theta}W_{old} + \frac{\sin(\tau\theta)}{\sin\theta}W_{new}, \ \ \tau \in [0,1] \tag{3}$$

where $\theta$ is the angle between the vectorized weights of ($W_{old}$ and $W_{new}$).

The core of MC-SMERG is to empirically determine the optimal fusion coefficient $\tau^*$ that balances past and current knowledge. In our exemplar-free setting, we construct a small virtual validation set by sampling features from stored class statistics ($\mu_c, \Sigma_c$) of all previously seen classes. We then conduct a lightweight grid search over $\tau$ (*e.g.*, 9 steps between 0 and 1). For each candidate $\tau$, we instantiate the interpolated adapter $W(\tau)$, evaluate cross-entropy loss on the virtual validation set, and select $\tau^*$ yielding the lowest loss. The final fused adapter is updated as $W_{fused} = W(\tau^*)$. This procedure enables MC-SMERG to locate a sweet spot along the low-loss path, integrating new knowledge while mitigating forgetting, without storing real exemplars or requiring expensive computation, as illustrated in Figure 6. Notably, this search for the optimal merge coefficient is highly efficient, taking only 8.3 ms on average per task in our experiments.

## 4 EXPERIMENTS

### 4.1 EXPERIMENTAL SETTING

**Datasets.** We test our approach on ImageNetSubset (100 classes from ImageNet (Russakovsky et al., 2015)) and CIFAR100 (Krizhevsky et al., 2009). To simulate LT-CIL scenarios, we construct CIFAR100-LT and ImageNetSubset-LT by applying an imbalance ratio $\rho$ following (Liu et al., 2022), with $\rho = 0.01$ to reflect more challenging settings. Also, we evaluate robustness under larger-scale conditions by extending the number of classes to 200 and 300 in the long-tailed scenario.

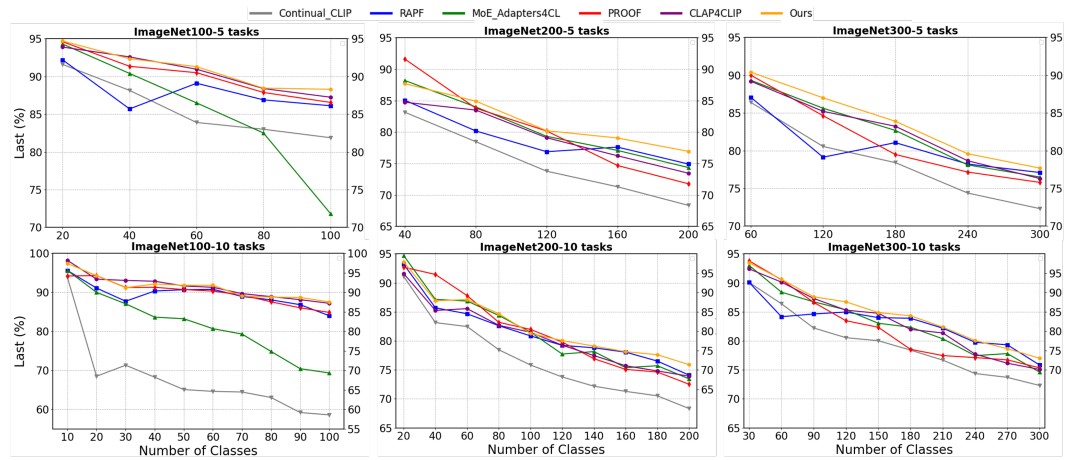

Figure 7: Performance comparison on ImageNetSubset-LT across various settings.

**Settings.** We follow the standard long-tailed class-incremental learning (LT-CIL) setup (Liu et al., 2022), but emphasize the more challenging shuffled scenario. Unlike the conventional ordered setting, where the class distribution remains consistent across tasks and difficulty increases predictably, the shuffled scenario imposes no assumptions on data distribution, yielding tasks with random and varying imbalance ratios. For all experiments, we report the standard CIL metrics: average incremental accuracy (Avg) and final accuracy (Last) after the last task. See Appendix for more detailed training and implementation settings.

**Baselines.** We benchmark our approach against state-of-the-art CLIP-based continual learning methods spanning diverse strategies. We compare with PROOF (Zhou et al., 2025), an exemplar-based method with parameter isolation; MoE-Adapters4CL (Yu et al., 2024), which combines a Mixture-of-Experts (MoE) with LoRA; CLAP4CLIP (Jha et al., 2024), which employs variational inference; RAPF (Huang et al., 2024), a statistics-based exemplar-free approach; and Continual-CLIP (Thengane et al., 2022), representing the baseline zero-shot CLIP performance. We use CLIP ViT-B/16 (Dosovitskiy et al., 2020) pre-trained by OpenAI (Radford et al., 2021) for all the methods.

### 4.2 ANALYSES

We present the main results of our method against state-of-the-art CLIP-based CIL baselines in Table 1. The experiments are conducted on ImageNetSubset-LT under increasingly challenging scenarios with 100, 200, and 300 total classes. The results clearly demonstrate that our proposed method consistently and significantly outperforms all baselines across all settings. For a more detailed view, Figure 7 visualize the accuracy trend as tasks progress. The graphs show that our method's performance curve remains higher than the baselines. We also evaluate our method on the conventional CIFAR100-LT benchmark. These detailed results are provided in Appendix.

Notably, as the number of classes increases and the task becomes more challenging, our method consistently maintains a large margin over baselines. In the 10-task setting, it surpasses the strongest exemplar-free baseline (RAPF) by 1.19%, 1.77%, and 1.68% in average accuracy for 100, 200, and 300 classes, respectively. More significantly, our exemplar-free approach even outperforms strong exemplar-based methods such as PROOF and MoE-Adapters4CL, with the advantage most pronounced in the 300-class scenario (Figure 7). These results highlight the scalability and robustness of our framework, demonstrating its effectiveness where the class space continually expands.

### 4.2.1 FINE-GRAINED ANALYSIS ON TAIL-CLASS PERFORMANCE

While our method demonstrates superior overall performance in Table 1, a crucial question remains: is this improvement specifically driven by our proposed tail-focused mechanisms? To answer this, we conduct a fine-grained analysis of the final accuracy on different class groups. Following the standard protocol for long-tail recognition analysis, we partition the classes of each benchmark into three groups based on the number of training samples: 'Low' (classes with the fewest samples), 'Mid', and 'High' (classes with the most samples). As shown in Tables 2 and 3, our method

| Method | Low | Mid | High | Last |
|---|---|---|---|---|
| Continual-CLIP | 82.94 | 81.21 | 81.45 | 81.88 |
| RAPF | 80.29 | 89.55 | 84.48 | 84.02 |
| MoE-Adapters4CL | 65.88 | 72.48 | 70.42 | 69.34 |
| Ours($\tau = 9$) | **86.14** | 84.93 | 92.79 | **87.54** |

Table 2: ImageNetSubset-100 (10 tasks).

| Method | Low | Mid | High | Last |
|---|---|---|---|---|
| ContinualCLIP | 66.54 | 67.33 | 71.01 | 68.34 |
| RAPF | 73.90 | 76.48 | 72.76 | 74.10 |
| MoE-Adapter4CL | 71.20 | 75.13 | 73.71 | 73.46 |
| Ours($\tau = 9$) | **76.42** | 74.93 | 76.78 | **76.20** |

Table 3: ImageNetSubset-200 (10 tasks).

| Ablation | Last (%) |
|---|---|
| Baseline(Adapter + SG) | 72.16 |
| Baseline w/ MC-SMeRG | 75.92 |
| Baseline w/ MC-SMeRG w/ TSHA | 76.32 |
| Baseline w/ MC-SMeRG w/ TSHA w/ TSS | 76.61 |
| Baseline w/ MC-SMeRG w/ TSHA w/ TSS + $\mathcal{L}_{\text{AMHL}}$ | **76.64** |

Table 4: Module ablation. SG refers the Gaussian-based feature sampling for previous classes.

| Task | Class | w/o $\mathcal{L}_{\text{AMHL}}$ | w/ $\mathcal{L}_{\text{AMHL}}$ |
|---|---|---|---|
| Old | Coffee mug | 17 | 35 |
| | Pot | 24 | 30 |
| New | Tea pot | 46 | 47 |
| | Cup | 43 | 43 |

Table 5: Ablation on AMHL on a hard case inter-task confusion scenario.

achieves the highest accuracy not only in the overall 'Last' metric but, more importantly, in the most challenging 'Low' sample group for both ImageNetSubset-100 and -200. For instance, on ImageNetSubset-100, our method outperforms the strongest exemplar-free baseline (RAPF) by a significant margin of nearly 6% on the 'Low' group (86.14% vs. 80.29%). This result provides direct evidence that our state-of-the-art performance is primarily driven by our method's superior ability to learn from data-scarce tail classes, validating the effectiveness of our proposed components.

### 4.2.2 ABLATION STUDY

To validate the contribution of each component, we conduct an ablation study on ImageNetSubset-LT (200 classes, 5 tasks), incrementally adding each proposed module to a simple baseline. As shown in Table 4, each component provides a clear performance boost. MC-SMERG provides a significant initial gain over the baseline's simple fusion, while Two Stage Hybrid Feature Space Augmentation(TSHA) and TSS further improve performance by stabilizing tail-class representations. While the final inclusion of our adaptive margin loss($\mathcal{L}_{\text{AMHL}}$) provides a modest boost to the overall accuracy, its critical role is revealed in a more challenging, fine-grained analysis. To isolate the effect of AMHL, we designed a "hard case" scenario in Table 5. Here, we evaluate predictions on 50 test images for each of the past tail classes ('Coffee mug', 'Pot') after a new, semantically similar head class ('Tea pot') is learned. Without $\mathcal{L}_{\text{AMHL}}$, the model suffers from severe catastrophic forgetting on the old classes. With $\mathcal{L}_{\text{AMHL}}$, the performance on these vulnerable classes is improved, demonstrating its crucial effectiveness in preserving knowledge and preventing inter-task confusion under the most difficult conditions. This targeted analysis reveals that the overall performance gain from $\mathcal{L}_{\text{AMHL}}$ is driven by its powerful, protective effect on the classes most susceptible to being forgotten. Also, we analyze the sensitivity of our key hyperparameters of the proposed method: $K$ for TSS and the number of search steps $\tau$ for MC-SMERG, which is reasonably stable. The detailed illustration is given in the Appendix.

## 5 CONCLUSION

In this paper, we address the challenging problem of exemplar-free, long-tailed class-incremental learning, a realistic scenario where the potential of pre-trained models like CLIP is largely untapped. Our framework unleashes CLIP's capabilities via a single lightweight adapter and a suite of synergistic techniques: Two-Stage Hybrid Augmentation, Tail-aware Semantic Shrinkage, Adaptive Margin Hinge Loss, and Mode-Connectivity Spherical Merge that work in concert to regularize the feature space and consolidate knowledge. Extensive experiments on the challenging ImageNetSubset-LT benchmark validate our approach. While the absolute performance gains may appear modest, they are highly significant within the severe constraints of this exemplar-free setting. Crucially, our method proves its superior scalability and stability, as the performance gap over baselines consistently widens as the number of classes increases. We show that leveraging CLIP's rich prior knowledge with minimal modification provides a robust and efficient path toward practical, scaled-up continual learning systems.

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

# A    APPENDIX

## A.1    CLASSES OF IMAGENETSUBSET

For our ImageNet100, ImageNet200, and ImageNet300 benchmarks, we selected subsets of categories from the full ImageNet1k dataset.

**Classes of ImageNet100**    To ensure fair comparisons across the 100-class subset, we strictly adhered to the class list and long-tail distribution settings proposed by prior research Liu et al. (2022).

**Classes of ImageNet200**    The selection of classes for the ImageNet200 subset was inspired by the class distribution observed in the VizWiz-Classification dataset Bafghi & Gurari (2023). The VizWiz dataset primarily contains images captured by blind and low-vision users, often featuring object classes distinct from those typically highlighted in standard benchmarks. By referencing this unique distribution, we aimed to create a challenging and diverse benchmark that effectively evaluates the scalability and robustness of class incremental learning (CIL) methods. The specific class names included in our ImageNet200 subset are listed below:

tiger, ice bear, hog, lion, umbrella, soccer ball, grand piano, laptop, strawberry, shopping cart, racer, sports car, pickup, trailer truck, bookcase, china cabinet, chiffonier, table lamp, file, folding chair, rocking chair, studio couch, toilet seat, desk, wardrobe, Granny Smith apple, orange, lemon, pineapple, banana, pomegranate, corn, upright, chime, acoustic guitar, electric guitar, valley, can opener, screwdriver, loudspeaker, microphone, mouse, electric fan, strainer, space heater, stove, rule, scale, analog clock, digital clock, wall clock, digital watch, projector, sunglasses, computer keyboard, lighter, desktop computer, printer, vending machine, joystick, hook, car wheel, swing, radiator, car mirror, remote control, buckle, seat belt, candle, fly, bee, iron, espresso maker, microwave, toaster, vacuum, dishwasher, refrigerator, washer, Crock Pot, frying pan, wok, teapot, patio, library, restaurant, grocery store, fountain, chainlink fence, picket fence, sliding door, pedestal, mashed potato, bell pepper, broccoli, cauliflower, cucumber, artichoke, mushroom, shower curtain, jean, carton, handkerchief, sandal, ashcan, plate, necklace, pajama, running shoe, chest, modem, tub, tray, book jacket, beer bottle, tile roof, teddy, pop bottle, suit, red wine, Christmas stocking, menu, stage, baseball, sunscreen, lampshade, bow tie, water jug, bucket, soup bowl, chain, mixing bowl, wine bottle, binder, cardigan, sweatshirt, pot, hamper, backpack, cup, radio, dough, lipstick, monitor, vase, paper towel, envelope, perfume, bathtub, hotdog, pillow, toilet paper, cassette, lotion, hair spray, pill bottle, window shade, barrel, washbasin, ballpoint pen, basketball, bath towel, window screen, cellular telephone, mailbox, fire screen, packet, pole, wallet, cassette player, comic book, piggy bank, street sign, Windsor tie, purse, television, measuring cup, espresso, pizza, wooden spoon, saltshaker, ball player, goblet, water bottle, soap dispenser, T-shirt, plastic bag, diaper, doormat, Loafer, ice cream, pretzel, quilt, tape player, iPod, pitcher, sock, CD player, cheeseburger, coffee mug.

**Classes of ImageNet300**    For the 300 class subset, we combined classes from the 100-class and 200-class subsets. To resolve the overlap of 23 duplicated classes between these subsets, we randomly selected alternative classes to replace them, maintaining a distinct and representative set. The specific class names are listed below:

tench, goldfish, great white shark, tiger shark, hammerhead shark, electric ray, stingray, hen, ostrich, brambling, house finch, junco, American robin, bulbul, jay, magpie, chickadee, American dipper, kite (bird of prey), bald eagle, vulture, great grey owl, fire salamander

# B    TRAINING DETAILS.

We use PyTorch 2.5.1 and experiments were conducted on an RTX 4070 ti super GPU. Most implementation details follow those of the prior work Huang et al. (2024). For our proposed components, we set the semantic neighbor numbers as $K = 3$ for TSS, the adaptive margins for AMHL to 0.2 (tail) and 0.1 (head), and the number of grid search steps for MC-SMERG to 9, for all the evaluations in the section.

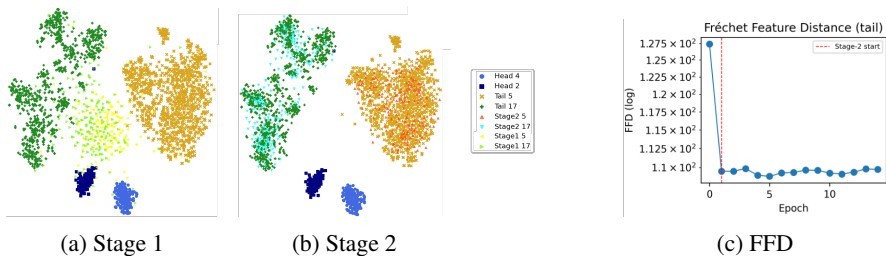

(a) Stage 1                 (b) Stage 2                          (c) FFD

Figure 8: Qualitative effect of the proposed Two-Stage Hybrid Feature Space Augmentation.

## B.1 QUALITATIVE EVIDENCE FOR TWO-STAGE HYBRID FEATURE SPACE AUGMENTATION

We provide qualitative evidence for our two-stage augmentation strategy, illustrating how it progressively refines tail-class features to align with the real data distribution. As shown in Figure 8, this process is visualized using t-SNE plots and validated quantitatively with the Fréchet Feature Distance (FFD).

In Stage 1 (Figure 8a), initial features are generated around the text embedding, creating a stable "semantic landing pad" that avoids interfering with existing classes during the model's unstable early training phase. Once the adapter stabilizes, Stage 2 begins (Figure 8b). Features are now generated from a k-NN core set of real data and are absorbed into the true tail-class cluster, increasing its density.

The FFD curve in Figure 8c underscores the effectiveness of our two-stage strategy. The abrupt drop at the onset of Stage 2 reflects the immediate benefit of moving from text-based seeding to data-driven refinement, while the subsequent stabilization indicates that the synthetic distribution has converged to the real data distribution. These results confirm the necessity of our hybrid approach a safety-oriented, semantic initialization followed by adaptive, data-driven tuning.

## B.2 MORE RESULTS OF EXPERIMENTS

### B.2.1 RESULTS ON CIFAR-100

For a comprehensive comparison, we also evaluate our method on the conventional CIFAR100-LT benchmark. The results are summarized in Table 6. In the 5-task setting, our exemplar-free method outperforms all baselines, including strong exemplar-based approaches. In the 10-task setting, our method remains the top-performing exemplar-free approach and achieves performance highly competitive with the best exemplar-based method (MoE-Adapters4CL).

### B.2.2 TAIL CLASS ACCURACY

To provide a more fine-grained analysis of our method's performance, especially on the most challenging tail classes, we partition the classes of each benchmark into three groups based on the number of training samples. The 'Low' group contains the 1/3 of classes with the fewest samples, 'Mid' contains the middle 1/3, and 'High' contains the 1/3 with the most samples. We then evaluate the final accuracy for each group.

(a) Hyperparameter $\tau$      (b) Hyperparameter $K$

Figure 9: Hyperparameter Sensitivity Analysis. Performance by search steps($\tau$) for MC-SMERG and neighbors($K$) for TSS.

| Methods | Exemplar | CIFAR100-LT | | | |
|---------|----------|-------------|---|---|---|
| | | 5 | | 10 | |
| | | Last(%) | Avg(%) | Last(%) | Avg(%) |
| Continual-CLIP | ✗ | 68.93 | 75.72 | 68.93 | 72.35 |
| RAPF | ✗ | 78.60 | 85.58 | 77.94 | 84.80 |
| MoE-Adapters4CL | ✓ | 80.06 | 87.24 | **78.60** | **86.46** |
| PROOF | ✓ | 72.88 | 79.71 | 72.26 | 80.68 |
| CLA4CLIP | ✓ | 73.00 | 78.83 | 73.75 | 76.39 |
| Ours | ✗ | **80.67** | **87.44** | 78.18 | 86.27 |

Table 6: Comparison on CIFAR100-LT benchmarks with imbalance factor $\rho = 0.01$. '5' and '10' represent the number of tasks, and 'Last' and 'Avg' denote the last and average accuracy across tasks.

| Method | Low | Mid | High | Last |
|--------|-----|-----|------|------|
| Continual-CLIP | 72.60 | 74.28 | 59.10 | 77.28 |
| RAPF | 75.62 | 82.47 | 75.29 | 77.52 |
| MoE-Adapters4CL | 79.22 | 82.04 | 74.29 | 78.6 |
| Ours($\tau = 9$) | **79.48** | 82.00 | 73.12 | 78.18 |

Table 7: Per-group accuracy comparison on CIFAR-100 (10 tasks).

**Results on CIFAR-100.** The results on CIFAR-100, presented in Table 7, further highlight the unique strength of our framework. Notably, while our overall Last accuracy (78.18%) on this benchmark was slightly lower than the exemplar-based MoE-Adapters4CL (78.60%), our method achieves a significantly higher accuracy on the critical Low sample group (79.48% vs. 79.22%). This detailed analysis reveals that our method's strength lies in its superior ability to learn from the most data-scarce classes. This is a crucial finding, as it demonstrates that our exemplar-free approach is more effective at tackling the core challenge of the long-tail problem, even when compared to strong methods that rely on storing past data.

In summary, this per-group analysis consistently shows that our method's state-of-the-art performance is driven by its remarkable ability to improve accuracy on the most challenging tail classes, which is a direct result of our proposed tail-focused mechanisms.

### B.2.3 HYPER-PARAMETER SENSITIVITY

We analyze the sensitivity of our key hyperparameters in Figure 9. We vary the number of neighbors K for TSS and the number of search steps $\tau$ for MC-SMERG. The results show that our method's performance is stable across a reasonable range of hyperparameter values, peaking around our chosen configuration ($K = 3$, $\tau = 9$). This demonstrates the robustness of our framework and indicates that it does not require extensive, sensitive tuning.

**The use of LLMs.** We use LLMs only for minor language editing, including adjustments to word choices and clarity. LLMs played no role in the research design, analysis, interpretation, or manuscript preparation.

