# OpenReview forum: "Rethinking CLIP for Long-Tailed Class-Incremental Learning"
_ICLR.cc/2026/Conference — ICLR 2026 Conference Withdrawn Submission_

### Official Review · Reviewer_unRV · 2025-10-23

**Soundness:** 2
**Presentation:** 2
**Contribution:** 2
**Rating:** 2
**Confidence:** 4

**Summary:**

This paper addresses long-tailed class incremental learning (CIL) with pre-trained vision-language models like CLIP. The authors propose a unified framework with three stages: Intra-Task Stabilization (Two-Stage Hybrid Augmentation), Inter-Task Preservation (Tail-Aware Semantic Shrinkage and Adaptive Margin Hinge Loss), and Knowledge Consolidation (Mode-Connectivity Spherical Merge). Using only a single lightweight adapter, the method efficiently leverages CLIP’s priors to handle data imbalance and forgetting. Experiments on long-tailed ImageNetSubset and CIFAR-100 show clear improvements over prior CLIP-based CIL methods.

**Strengths:**

1. Clear three-stage design: Provides a structured and logical approach to long-tailed CIL.

2. Lightweight and efficient: Uses a single adapter while maintaining strong performance.

3. Extensive experiments: Demonstrates consistent gains and robustness across multiple long-tailed benchmarks.

**Weaknesses:**

1. The paper does not clearly justify the necessity of studying long-tailed class-incremental learning (CIL) in the context of large-scale pre-trained vision-language models like CLIP. The proposed problem setting appears more like a forced combination of two popular research topics without sufficient motivation or evidence that such integration introduces new challenges or insights.

2. The paragraphs on lines 104 and 135 in Section 2.2 are exactly repeated.

3. The two t-SNE visualizations in Figure 1 do not show clear qualitative differences. In fact, the class boundaries appear reasonably well separated, which contradicts the claim that they “appear diffuse and weakly separated.” The numerical differences reported in the accompanying table are also small, and the paper does not explain how these numbers are computed. Without this clarification, the visual and quantitative evidence does not convincingly support the authors’ argument.

4. Most of the proposed techniques rely on the assumption that data features follow a Gaussian distribution. This is an idealized and often unrealistic assumption for real-world visual data. The paper lacks discussion, justification, or empirical verification of this assumption, which weakens the theoretical soundness of the approach.

5. In Table 4, it is unclear why the “Last” metric is chosen as the main indicator for ablation studies. The authors should provide a stronger explanation for this choice. Moreover, the reported results show that the Two-Stage Hybrid Augmentation (TSS) brings only marginal improvements, which raises questions about its effectiveness and contribution.

6. The paper does not include any analysis or discussion of hyperparameters. Since the proposed framework contains several key parameters , a sensitivity study would be necessary to evaluate robustness and reproducibility.

**Questions:**

See weakness.

---

### Official Review · Reviewer_gRXA · 2025-10-24

**Soundness:** 2
**Presentation:** 2
**Contribution:** 2
**Rating:** 4
**Confidence:** 4

**Summary:**

This paper aims to address the challenging problem of exemplar-free, long-tailed class-incremental learning with CLIP model.  Specifically, this paper proposes a suite of synergistic techniques: (1) Two-Stage Hybrid Augmentation for robust tail-class feature augmentation; (2)Tail-aware Semantic Shrinkage to correct tail-class statistics; (3) Adaptive Margin Hinge Loss to refine decision boundaries;  (4)Mode-Connectivity Spherical Merge for stable integration of knowledge into the single adapter. Experiments on the challenging ImageNetSubset-LT benchmark validate the proposed approach.

**Strengths:**

1.	It is an interesting idea to fully utilize CLIP’s pretrained feature space to prevent class interference for class incremental learning.

2.  Experiments on the challenging ImageNetSubset-LT benchmark validate the proposed approach.

**Weaknesses:**

1.	The manuscript can be further improved. First, in section 3.1, although the author argues that RAPF fails in Figure1, it seems that figure 1(a) and 1(b) are quite similar. Why the author argues that RAPF fails in Figure1 needs further clarification. Similarly, Figure 2 doesn't clearly support the failure of Inter-Task Preservation and further discussion is needed. Second, while the paper conducted experiments on two datasets, only the results for one dataset are included in the main paper, which makes the empirical validation less convincing. Third, the discussion of related work is not sufficient. There is a vast amount of literature of class incremental learning, but the paper only discusses a few of them.

2.	The empirical validation is not sufficient. The paper conducted experiments on ImageNetSubset-LT and CIFAR100. Although the performance on ImageNetSubset dataset is better than baselines, the proposed method doesn’t outperform all the baselines on CIFAR100. Further empirical validation on more datasets is needed to support the advantage of the proposed method. Refer to [1] for more datasets (and/or exemplar-free CIL baselines) in CIL validation.

[1] External Knowledge Injection for CLIP-Based Class-Incremental Learning

**Questions:**

1. Duplicated paragraph(line 137), typo(figure 1 caption: class 38 or class 36), texts and legends need to be larger in figure 5(c), 1(a), 1(b)

2.	In section 3.2.2, why does the paper shrink the tail classes toward their semantically related head-class neighbors? How does the paper define and find the neighbors? Will the shrink lead to confusion in classification, as shrinking mixes up class statistics?

3.	What is the explicit definition of $f_{edge}$? What is the procedure to get it ?

4.	In Table 4, what is the main difference between “Baseline(Adapter + SG)” and “Baseline w/ MC-SMeRG”? Does “Baseline(Adapter + SG)” have a merge process?

---

### Official Review · Reviewer_Ei4N · 2025-10-29

**Soundness:** 2
**Presentation:** 2
**Contribution:** 2
**Rating:** 2
**Confidence:** 4

**Summary:**

This paper addresses the challenging problem of exemplar-free long-tailed class-incremental learning (LT-CIL), where models need to learn new classes sequentially under extreme data imbalance without storing historical samples. Existing CLIP-based CIL methods often degrade sharply in long-tailed scenarios due to insufficient utilization of pre-trained knowledge and failures in intra-task stabilization and inter-task preservation. To tackle these issues, the authors propose a lightweight framework with a single adapter, structured into three core stages: Intra-Task Stabilization via Two-Stage Hybrid Augmentation (TSHA) to refine tail-class features, Inter-Task Preservation using Tail-aware Semantic Shrinkage (TSS) and Adaptive Margin Hinge Loss (AMHL) to protect past knowledge and optimize decision boundaries, and Knowledge Consolidation through Mode-Connectivity Spherical Merge (MC-SMERG) to fuse task-specific adapters. Extensive experiments on ImageNetSubset-LT and CIFAR100-LT benchmarks demonstrate that the proposed method outperforms state-of-the-art approaches, especially on data-scarce tail classes, and maintains scalability as the number of classes increases.

**Strengths:**

- The framework uses only one lightweight adapter, avoiding complex add-on modules, and the three-stage components (TSHA, TSS, AMHL, MC-SMERG) complement each other to solve intra-task instability and inter-task forgetting simultaneously.
- The integration of mode connectivity theory and spherical interpolation for adapter fusion offers novel insights into knowledge consolidation, while the exemplar-free design avoids memory and privacy issues.

**Weaknesses:**

- Limited Generalization to Other Architectures: The method is exclusively evaluated on CLIP ViT-B/16; its compatibility and performance with other vision-language models or backbone architectures (e.g., ViT-L/14, Flamingo) remain untested.
- Sensitivity to Hyperparameters in Complex Scenarios: Although the ablation study shows stability for key hyperparameters (K=3 for TSS, τ=9 for MC-SMERG), the performance under extreme parameter variations or cross-dataset hyperparameter transfer is not thoroughly discussed.
- Lack of Analysis on Computational Overhead: While the adapter is lightweight, the two-stage augmentation and virtual validation set construction for MC-SMERG may introduce additional computational costs, which are not quantified or compared with baselines.
- Insufficient Discussion on Semantic Similarity Metrics: The paper uses CLIP text embeddings to measure class similarity for TSS and AMHL, but alternative metrics (e.g., visual feature similarity) are not explored, leaving room for further optimization.
- The figures and tables in the paper are confusing. There are some figures showing no difference between the proposed approaches and baselines.

**Questions:**

- Please make a more comprehensive analysis of intra-class variance, such as in Figure 1, which should be a statistic of multiple classes rather than analyzing a few categories.
- Can we compare the results with the examplar, because some other methods are designed with the examples.

---

### Official Review · Reviewer_B3BQ · 2025-10-31

**Soundness:** 2
**Presentation:** 3
**Contribution:** 3
**Rating:** 4
**Confidence:** 4

**Summary:**

This submitted paper addresses the performance degradation of existing CLIP-based methods in long-tailed class-incremental learning (long-tailed CIL) and proposes an exemplar-free framework relying solely on a single lightweight adapter. First, the authors conduct the first in-depth study on the limitations of modern CLIP-based CIL methods in realistic long-tailed scenarios, highlighting the need to maximize the utilization of pre-trained knowledge rather than relying on complex add-on modules. Second, the framework solves core challenges through a synergistic three-stage design: in the Intra-Task Stabilization stage, Two-Stage Hybrid Feature Augmentation (initialization based on text priors and k-NN data-driven refinement) enhances the robustness of tail-class representations; in the Inter-Task Preservation stage, Tail-aware Semantic Shrinkage (correcting tail-class statistical bias using head classes) and Adaptive Margin Hinge Loss (dynamically adjusting boundaries to protect tail classes) alleviate forgetting and inter-class confusion; in the Knowledge Consolidation stage, Mode-Connectivity Spherical Merge (SLERP interpolation and virtual validation set optimization) realizes stable unification of old and new adapters. Finally, experiments on ImageNetSubset-LT (100/200/300 classes) and CIFAR100-LT show that the method consistently outperforms existing SOTA approaches (including exemplar-free and exemplar-based methods), with the performance advantage expanding as the number of classes increases (more severe long-tail conditions), verifying its scalability and efficient learning ability for tail classes, and providing an effective path for practical incremental learning systems.

**Strengths:**

1. The paper proposes a three-stage framework that systematically addresses instability and forgetting issues in long-tailed class-incremental learning.
2. The authors demonstrate that using a single shared lightweight adapter is sufficient to achieve significant performance gains, showing better practicality and scalability.
3. The two key components, TSHA and TSS, are specifically designed to tackle long-tailed scenarios, effectively mitigating the degradation of tail-class representations.

**Weaknesses:**

1. Although the overall framework is well-structured, most components are combinations or refinements of existing ideas, lacking strong novelty or justification.
2. The paper does not provide an explanation for why interpolation along a low-loss path in the weight space used in MC-SMERG can preserve old knowledge and prevent forgetting.
3. While the authors emphasize that the method is lightweight and efficient, the paper only reports the adapter fusion time (8.3 ms per task) and lacks a detailed comparison of total training time, GPU memory usage, and parameter size against baselines.

**Questions:**

How sensitive is the approach to the imbalance ratio ρ and task ordering?

Other questions see weakness.

---

### Note · Authors · 2026-01-15

I have read and agree with the venue's withdrawal policy on behalf of myself and my co-authors.